# The association between attachment pattern and depression severity in Thai depressed patients

**Chotiman Chinvararak**[1]*, **Pantri Kirdchok**[1], **Peeraphon Lueboonthavatchai**[2]

**1** Department of Psychiatry, Faculty of Medicine Vajira Hospital, Navamindradhiraj University, Bangkok, Thailand, **2** Department of Psychiatry, Faculty of Medicine, Chulalongkorn University, Bangkok, Thailand

* chotiman@nmu.ac.th

## Abstract

### Objective

We aimed to study attachment patterns and their association with depression severity in Thai depressed patients.

### Method

We conducted a descriptive study of depressed participants at King Chulalongkorn Memorial Hospital from November 2013 to April 2014. The Thai Short Version of Revised Experience of Close Relationships Questionnaire and the Beck Depression Inventory-II (BDI-II) were administered to all participants. We assessed BDI-II scores, classified by attachment patterns, using one-way analyses of variance. The associated factors and predictors of depression severity were analysed by chi-square and logistic regression analyses, respectively.

### Results

A total of 180 participants (75% female; mean age = 45.2 ± 14.3 years) were recruited. Dismissing attachment was the most common pattern in Thai depressed patients (36.1%). Depressed patients with preoccupied attachment demonstrated the highest BDI-II scores. The best predictor of moderate to severe depression severity was preoccupied/fearful attachment (odds ratio = 3.68; 95% confidence interval = 2.05–7.30).

### Conclusions

Anxious attachment was found to be associated with higher depression severity. Preoccupied/fearful attachment was the predictor of moderate to severe depression severity.

**Data Availability Statement:** All relevant data are within the paper and its Supporting Information files.

**Funding:** The authors received no funding for this work.

**Competing interests:** The authors have declared that no competing interests exist.

## Introduction

Major depressive disorder is one of the most common psychiatric disorders, and is the third leading cause of disability-adjusted life-years [1]. While the worldwide prevalence of depression is between 4.4–10.8%, in Thailand it is between 2.4–3.2% [1–3]. Severe depression leads to impaired daily life function, low productivity, and suicides, and has been found to be related with sex, age, education, comorbid anxiety symptoms, medical illnesses including metabolic disorders, substance use, social support, and stressful life events [1, 4–10].

Attachment is developed from infancy and persists across the lifespan [11]. Adult attachment patterns are classified into four subtypes: secure, preoccupied, fearful, and dismissing [12], with the latter three demonstrating insecure attachment. Adults with insecure preoccupied or fearful attachment styles have been found to have higher incidences of mental health conditions [13, 14]. While the pathophysiology of depression is still not clear [15], attachment theory is one of the developmental theories widely used to explain the psychological aetiology of psychiatric disorders including depressive disorder [14, 16–18].

The objective of the present study was to investigate the attachment pattern and its association with depression severity in Thai patients with major depressive disorder. We hypothesize that depressed patients with anxious attachment style are likely to have a higher degree of depression compared with those with secure attachment; however, there are limited studies concerning the role of attachment and depression severity in depressed patients, particularly in Thailand. Studying and understanding attachment patterns will assist clinicians in providing appropriate care to depressed patients and promoting mental well-being.

## Materials and methods

### Design, settings, and study sample

We conducted a cross-sectional descriptive study following STROBE guidelines [19]. As the proportion(p) of moderate to severe depression was 0.41, sample size was estimated by p = 0.5. Using alpha at 0.05 and power at 0.9, the required sample size was 93 [5, 20]. One hundred and eighty depressed participants aged 18 years and older were recruited by purposive sampling from the Department of Psychiatry at King Chulalongkorn Memorial Hospital in Bangkok from November 2013 to April 2014. We obtained approval from the Ethical Committee of the Institutional Review Board of the Faculty of Medicine at Chulalongkorn University (COA no. 687/2013). Participants were required to be diagnosed with major depressive disorder by the Diagnostic and Statistical Manual of Mental Disorders, 5th edition, and were excluded if they had any recorded active medical conditions or other major psychiatric disorders over the previous month [21]. Those who met the eligibility criteria were informed of the study's objectives and method and provided written informed consent.

### Data collection

All participants completed the following questionnaires: a demographic data form, the Thai Short Version of Revised Experience of Close Relationships Questionnaire (ECR-R-18), and the Beck Depression Inventory II (BDI-II).

The Thai ECR-R-18 was used to measure attachment patterns [22]. It consists of 18 questions divided into anxiety and avoidance dimensions. The cut-off value of each dimension at $\geq 4$ points indicates high levels of anxiety or avoidance. The attachment pattern can then be classified as secure (low anxiety, low avoidance), preoccupied (high anxiety, low avoidance), fearful (high anxiety, high avoidance), or dismissing (low anxiety, high avoidance). We

considered preoccupied and fearful pattern as an anxious attachment. By contrast, secure and dismissing patterns were categorized as a non-anxious attachment [12].

The BDI-II, a widely used questionnaire to assess depression severity, consists of 21 questions with a total possible score of 63 [23]. The severity of depression can be categorized as minimal (0–13), mild (14–19), moderate (20–28), and severe depression (29–63).

## Statistical analyses

Data were analysed using SPSS software (version 22.0; IBM, Chicago, IL, USA). The attachment pattern is presented by frequency and percentage. One-way analyses of variance were used to compare BDI-II scores classified by attachment pattern. The associated factors of depression severity were analysed by chi-square test. Significant factors from the theoretical review [1, 4–10] and univariate analysis were entered into multiple logistic regression models (odds ratio [OR] and 95% confidence interval [CI]) to identify potential predictors of depression severity. $P < 0.05$ was considered statistically significant.

## Results and discussion

We recruited a total of 180 participants (mean age = 45.2 ± 14.3 years). Most participants were female (75.0%), married (43.9%), had a bachelor's degree (38.9%), and had adequate income (77.8%). Approximately 66% of participants had at least one physical illness. Roughly 33% of participants had a history of substance use within the last year. Finally, 88.9% of participants were prescribed antidepressants and 16.1% had a history of psychiatric hospitalization Table 1.

**Table 1. Participant's characteristics.**

| Characteristics | N (%) or Mean±SD | Characteristics | N (%) or Mean±SD |
|---|---|---|---|
| Sex | | History of medical illness | 113 (62.8) |
| Female | 135 (75.0) | Common medical illness | |
| Male | 45 (25.0) | Hyperlipidemia | 46 (25.6) |
| Age (years) | 45.2±14.3 | Hypertension | 41 (22.8) |
| min = 18 max = 83 | | Musculoskeletal | 34 (18.9) |
| Marital status | | disorders | |
| Single | 74 (41.1) | Allergy | 27 (15.0) |
| Married | 79 (43.9) | Gastrointestinal tract disorders | 25 (13.9) |
| Widow | 14 (7.8) | Diabetes | 14 (7.8) |
| Divorce or | 13 (7.2) | History of Substances Use | 58 (32.2) |
| Separation | | (within 1 year) | |
| Education | | Alcohol | 31 (17.2) |
| Primary school | 35 (19.4) | Tobacco | 13 (7.2) |
| Middle school | 23 (12.7) | Others | 4 (2.2) |
| High school | 20 (11.1) | Psychotropic drugs | |
| Diploma | 12 (6.7) | Antidepressants | 160 (88.9) |
| Bachelor | 70 (38.9) | Benzodiazepines | 91 (50.6) |
| Higher than | 20 (11.1) | Antipsychotics | 22 (12.2) |
| Bachelor | | Mood stabilizers | 5 (2.8) |
| Income | 637.8 | History of Psychiatric | 29 (16.1) |
| (USD/month) median (IQR) | (318.9–956.6) | Hospitalization | |
| Adequate income | 140 (77.8) | | |

Abbreviation: IQR, interquartile range.

**Table 2. Attachment pattern and BDI-II score.**

| Attachment pattern | n (%) | BDI-II score (Mean±SD) |
|---|---|---|
| Secure | 44 (24.4) | 18.36±13.30 |
| Preoccupied | 28 (15.6) | 26.68±10.54 |
| Fearful | 43 (23.9) | 22.84±13.62 |
| Dismissing | 65 (36.1) | 12.91±11.46 |

Almost half of the participants were diagnosed with the minimum severity of depression (43.9%), followed by severe (22.8%), moderate (18.3%), and mild (15.0%). Dismissing attachment was the most common pattern found in these participants (36.1%), followed by secure (24.4%), fearful (23.9%), and preoccupied attachments (15.6%) Table 2. Depressed patients with preoccupied attachment demonstrated the highest BDI-II score, whereas those with dismissing attachment had the lowest score Tables 2 and 3.

The most substantial associated factor of depression severity was attachment pattern ($P < 0.01$) Table 4. The logistic regression analysis found that anxious attachment style (preoccupied/fearful) was the most statistically significant predictor for moderate to severe depression severity Table 5. In addition, AUCROC showed the value for anxious attachment style in predicting moderate to severe depression was 0.66 (Fig 1).

The present study found that attachment patterns associated with high levels of anxiety, namely preoccupied and fearful, were significantly predictive of moderate to severe depression severity. In addition, these patterns increased the likelihood of moderate to severe depression, after adjusting for sex, education, history of medical illness, and history psychiatric hospitalization (OR = 3.86; 95% CI: 2.05–7.30). The results from this study were consistent with prior studies that insecure attachment especially anxious attachment was correlated with severe depressive symptoms [24, 25]. Zhou et al. (2021) also found that comorbid anxiety symptoms were associated with suicidal attempts in major depressive disorder patients [4].

Attachment is a social connection formed between an infant and their primary caregiver [11]. A significant caregiver is important for emotional support during this critical period [11]. Inappropriate emotional support or adverse childhood experiences may cause insecure attachment styles [11, 26–28]. Many studies have found that attachment pattern is likely to persist into adulthood, as they will use their attachment style to relate to others [11, 12, 17, 18].

Anxious attachment patterns, including preoccupied and fearful, are associated with more severe depression [24, 25]. Although the exact pathophysiology of mental disorders remains unclear [15], psychiatrists believe that overall psychology is an essential component [29].

**Table 3. Compared BDI-II score by one-way ANOVA.**

| | Sum of Squares | Df | Mean Square | F | P-value |
|---|---|---|---|---|---|
| Between groups | 4703.649 | 3 | 1567.883 | 10.30 | <0.001** |
| Within groups | 26797.596 | 176 | 152.259 | | |
| Total | 31501.244 | 179 | | | |

*$P <0.05$, **$P <0.01$

Post Hoc Tests (Bonferroni)

| Attachment pattern | Secure | Preoccupied | Fearful | Dismissing |
|---|---|---|---|---|
| Secure | - | -8.32* | -4.47 | 5.46 |
| Preoccupied | 8.32* | - | 3.84 | 13.77* |
| Fearful | 4.47 | -3.84 | - | 9.93* |
| Dismissing | -5.46 | -13.77* | -9.93* | - |

*$P <0.05$

**Table 4. Factors associated with depression severity.**

| Variables | Depression Severity | | | | $\chi^2$ | P-value |
|---|---|---|---|---|---|---|
| | minimal to mild (n = 106) | | moderate to severe (n = 74) | | | |
| | N | % | N | % | | |
| Sex | | | | | | |
| Male | 29 | 64.4 | 16 | 35.6 | 0.8 | 0.38 |
| Female | 77 | 57.0 | 58 | 43.0 | | |
| Age (years) | | | | | | |
| 40 or lower | 34 | 50.0 | 34 | 50.0 | 3.6 | 0.06 |
| Higher than 40 | 72 | 64.3 | 40 | 35.7 | | |
| Education | | | | | | |
| Lower than bachelor | 57 | 52.2 | 43 | 47.8 | 3.3 | 0.07 |
| Bachelor or higher | 49 | 65.6 | 31 | 34.4 | | |
| Adequacy of income | | | | | | |
| Adequate | 83 | 59.3 | 57 | 40.7 | 0.1 | 0.75 |
| Inadequate | 23 | 57.5 | 17 | 42.5 | | |
| History of medical illness | | | | | | |
| Yes | 65 | 57.5 | 48 | 42.5 | 0.2 | 0.63 |
| No | 41 | 61.2 | 26 | 38.8 | | |
| History of substances use | | | | | | |
| Yes | 30 | 51.7 | 28 | 48.3 | 1.8 | 0.18 |
| No | 76 | 62.3 | 46 | 37.7 | | |
| History of psychiatric hospitalization | | | | | | |
| Yes | 16 | 59.6 | 13 | 40.4 | 0.2 | 0.66 |
| No | 90 | 55.2 | 61 | 44.8 | | |
| Attachment pattern (1) | | | | | | |
| Secure | 27 | 61.4 | 17 | 38.6 | 23.7 | <0.001** |
| Preoccupied | 8 | 28.6 | 20 | 71.4 | | |
| Dismissing | 20 | 46.5 | 23 | 53.5 | | |
| Fearful | 51 | 78.5 | 14 | 21.5 | | |
| Attachment pattern (2) | | | | | | |
| Low level of anxiety (secure/dismissing) | 28 | 39.4 | 43 | 60.6 | 18.3 | <0.001** |
| High level of anxiety (preoccupied/fearful) | 78 | 71.6 | 31 | 28.4 | | |

**P<0.01

Deficits in mentalization processes are an important risk factor for psychiatric disorders [13, 14, 16] because of difficulties in emotional regulation. Ciechanowski et al. (2002) determined that individuals with anxious attachment styles likely demonstrate ineffective communication skills, causing difficulties in the effective handling of psychosocial problems [18]. Edelstein and Shaver (2004) explained that individuals with high attachment anxiety often worry with their

**Table 5. Stepwise multiple logistic regression.**

| Variables | Adjusted OR | 95% CI of Adjusted OR | | P-value |
|---|---|---|---|---|
| | | Lower | Upper | |
| Anxious attachment (preoccupied/fearful attachment) | 3.86 | 2.05 | 7.30 | <0.001** |

**P<0.01, adjusted for sex, education, history of medical illness, and history of psychiatric hospitalization

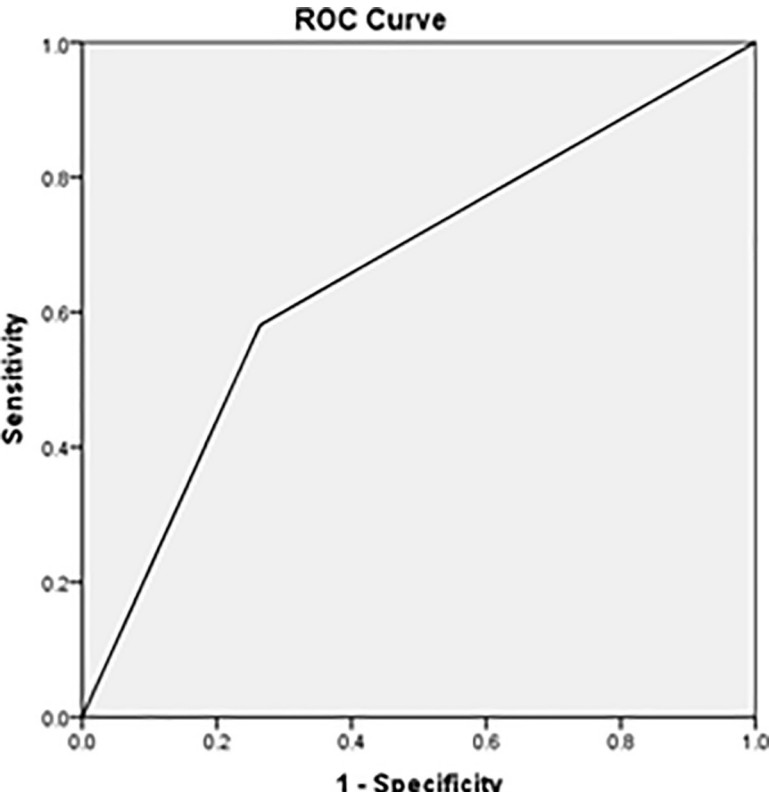

**Fig 1. The discriminatory capacity of anxious attachment style in predicting moderate to severe depression.** The area under the curve of preoccupied/fearful attachment was 0.66.

symptoms; consequently, they seek reassurance from medical professionals, which may disrupt a healthy doctor-patient relationship [30]. Contrastingly, those with dismissing style of attachment are likely to underreport their symptoms [18, 30].

We are aware of several limitations of the present study. First, due to the descriptive design, we can only indicate associated factors, not causal relationships. Secondly, most of our participants were female. Finally, we only collected samples from the Department of Psychiatry at King Chulalongkorn Memorial Hospital, which may not be representative of all depressed patients in other cultural settings.

According to present study, insecure attachment was commonly found in patients with major depressive disorder. More severe depression was associated with anxious attachment patterns. Intervention to promote secure attachment may be an important strategy to reduce the risk of severe depression into adulthood.

## Conclusions

Anxious attachment was the most common pattern in depressed patients and was associated with more severe depression. Understanding attachment patterns may be helpful for clinicians to develop and provide improved treatment to depressed patients.

## Supporting information

**S1 Data.**
(SAV)

## Acknowledgments

We would like to thank Professor Nuntika Thavichachart and Professor Tinakorn Wongpakaran for allowing us to use Thai versions of the Beck Depression Inventory-II and Short Version of Revised Experience of Close Relationships Questionnaire, respectively.

## Author Contributions

**Conceptualization:** Chotiman Chinvararak, Peeraphon Lueboonthavatchai.

**Data curation:** Chotiman Chinvararak, Pantri Kirdchok.

**Formal analysis:** Chotiman Chinvararak, Pantri Kirdchok, Peeraphon Lueboonthavatchai.

**Investigation:** Chotiman Chinvararak.

**Methodology:** Chotiman Chinvararak, Peeraphon Lueboonthavatchai.

**Project administration:** Chotiman Chinvararak.

**Software:** Chotiman Chinvararak.

**Supervision:** Peeraphon Lueboonthavatchai.

**Validation:** Chotiman Chinvararak, Pantri Kirdchok, Peeraphon Lueboonthavatchai.

**Writing – original draft:** Chotiman Chinvararak.

**Writing – review & editing:** Chotiman Chinvararak, Pantri Kirdchok, Peeraphon Lueboonthavatchai.

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
