## [Decision Letter · Decision Letter 0]

30 Jun 2021

PONE-D-21-10734

The association between attachment pattern and depression severity in Thai depressed patients

PLOS ONE

Dear Dr. Chinvararak,

Thank you for submitting your manuscript to PLOS ONE. After careful consideration, we feel that it has merit but does not fully meet PLOS ONE’s publication criteria as it currently stands. Therefore, we invite you to submit a revised version of the manuscript that addresses the points raised during the review process.

We look forward to receiving your revised manuscript.

Kind regards,

Zezhi Li, Ph.D., M.D.

Academic Editor

PLOS ONE

Additional Editor Comments:

Comments of reviewer 2:

This is a cross-section study describing the attachment pattern and depression severity in Thai depressed patients. It includes a good number of participants. The methods are appropriate, and the conclusion is clear. However, several minor issues need to be addressed:

1. An AUC-ROC curve would be helpful to demonstrate the capacity of preoccupied/fearful attachment in predicting depression.

2. It would be interesting to know the proportion of the participants with suicide attempts in Thai depressed patients. A pioneering study (listed below) has demonstrated that comorbid anxiety and metabolic abnormalities are also associated with severe major depressive disorders (MDD) with suicide attempts. Please include this in the discussion.

The association of clinical correlates, metabolic parameters, and thyroid hormones with suicide attempts in first-episode and drug-naïve patients with major depressive disorder comorbid with anxiety: a large-scale cross-sectional study. Transl Psychiatry. 2021 Feb 4;11(1):97. doi: 10.1038/s41398-021-01234-9. PMID: 33542178; PMCID: PMC7862235.

Journal Requirements:

"No"

"No"

Reviewers' comments:

Reviewer's Responses to Questions

**Comments to the Author**

1. Is the manuscript technically sound, and do the data support the conclusions?

Reviewer #1: Yes

Reviewer #2: Yes

2. Has the statistical analysis been performed appropriately and rigorously? 

Reviewer #1: I Don't Know

Reviewer #2: Yes

3. Have the authors made all data underlying the findings in their manuscript fully available?

Reviewer #1: Yes

Reviewer #2: Yes

4. Is the manuscript presented in an intelligible fashion and written in standard English?

Reviewer #1: Yes

Reviewer #2: Yes

5. Review Comments to the Author

Reviewer #1: Q1: With regard to the introduction of MDD, your research can refer to some newer literature, such as “Major Depressive Disorder: Advances in Neuroscience Research and Translational Applications.Li Z, Ruan M, Chen J, Fang Y.Neurosci Bull. 2021 Feb 13”.

Q2: In the methods section, the authors doesn’t mention the method to estimate sample size. Have you estimated the sample size in your study, and what is the specific method?

Q3: In the results section, your work lists a lot of participant’s characteristics about the enrolled patients, but are there differences in these characteristics among people with different attachment patterns? If so, when comparing the score difference of BDI-II between different attachment patterns, it may be a better choice to use the differences in participant’s characteristics as covariates to perform a covariance analysis.

Q4: In the results section, are there some problems with the layout of Table 4?

Q5: In the results section, if “Secure/Dismissing and Preoccupied/Fearful” layout in Table 4 means the number of Secure attachment pattern plus the number of Dismissing attachment pattern and the number of Preoccupied attachment pattern plus the number of Fearful attachment pattern and what is the basis for such a combination?

Q6: In the results section, your research adjusted sex, education, history of medical illness and history of psychiatric hospitalization to run stepwise multiple logistic regression, what’s your reason to adjust these variables?

6. PLOS authors have the option to publish the peer review history of their article (what does this mean?). If published, this will include your full peer review and any attached files.

Reviewer #1: No

Reviewer #2: No

---

## [Author Response · Author response to Decision Letter 0]

19 Jul 2021

We have attached the rebuttal letter to respond reviewers and editor comments.

---

## [Decision Letter · Decision Letter 1]

28 Jul 2021

The association between attachment pattern and depression severity in Thai depressed patients

PONE-D-21-10734R1

Dear Dr. Chinvararak,

We’re pleased to inform you that your manuscript has been judged scientifically suitable for publication and will be formally accepted for publication once it meets all outstanding technical requirements.

Kind regards,

Zezhi Li, Ph.D., M.D.

Academic Editor

PLOS ONE

Additional Editor Comments (optional):

Reviewers' comments:

Reviewer's Responses to Questions

**Comments to the Author**

1. If the authors have adequately addressed your comments raised in a previous round of review and you feel that this manuscript is now acceptable for publication, you may indicate that here to bypass the “Comments to the Author” section, enter your conflict of interest statement in the “Confidential to Editor” section, and submit your "Accept" recommendation.

Reviewer #1: All comments have been addressed

Reviewer #2: All comments have been addressed

2. Is the manuscript technically sound, and do the data support the conclusions?

Reviewer #1: Yes

Reviewer #2: Yes

3. Has the statistical analysis been performed appropriately and rigorously? 

Reviewer #1: Yes

Reviewer #2: Yes

4. Have the authors made all data underlying the findings in their manuscript fully available?

Reviewer #1: Yes

Reviewer #2: Yes

5. Is the manuscript presented in an intelligible fashion and written in standard English?

Reviewer #1: Yes

Reviewer #2: Yes

6. Review Comments to the Author

Reviewer #1: (No Response)

Reviewer #2: The authors have addressed all the comments in an appropriate manner and it is now suitable for publication.

7. PLOS authors have the option to publish the peer review history of their article (what does this mean?). If published, this will include your full peer review and any attached files.

Reviewer #1: No

Reviewer #2: No

---

## [Editor Report · Acceptance letter]

4 Aug 2021

PONE-D-21-10734R1 

The association between attachment pattern and depression severity in Thai depressed patients 

Dear Dr. Chinvararak:

I'm pleased to inform you that your manuscript has been deemed suitable for publication in PLOS ONE. Congratulations! Your manuscript is now with our production department. 

Kind regards, 

on behalf of

Dr. Zezhi Li 

Academic Editor

PLOS ONE